# Extended or Continuous Infusion of Carbapenems in Children with Severe Infections: A Systematic Review and Narrative Synthesis

**DOI:** 10.3390/antibiotics10091088

**Published:** 2021-09-09

**Authors:** Pengxiang Zhou, Yahui Zhang, Zhenhuan Wang, Yingqiu Ying, Yan Xing, Xiaomei Tong, Suodi Zhai

**Affiliations:** 1Department of Pharmacy, Peking University Third Hospital, Beijing 100191, China; pxzhou0427@bjmu.edu.cn (P.Z.); 1510307226@pku.edu.cn (Y.Z.); yingyingqiu@bjmu.edu.cn (Y.Y.); 2Institute for Drug Evaluation, Peking University Health Science Center, Beijing 100191, China; 3Department of Pharmacy Administration and Clinical Pharmacy, School of Pharmaceutical Sciences, Peking University, Beijing 100191, China; pwbwzh@sina.com; 4Department of Pharmacy, First Hospital of Tsinghua University, Beijing 100016, China; 5Department of Pediatrics, Peking University Third Hospital, Beijing 100191, China; xingyan@bjmu.edu.cn

**Keywords:** systematic review, narrative synthesis, pharmacokinetic, pharmacodynamic, carbapenems, meropenem, continuous infusion, extended infusion, short-term infusion, children, severe infections

## Abstract

We systematically reviewed the efficacy and safety of an extended or continuous infusion (EI/CI) versus short-term infusion (STI) of carbapenems in children with severe infections. Databases, including PubMed, Embase, the Cochrane Library, Clinicaltrials.gov, China National Knowledge Infrastructure, WanFang Data, and SinoMed, were systematically searched from their inceptions to 10 August 2020, for all types of studies (such as randomized controlled trials (RCTs), retrospective studies, and pharmacokinetic or population pharmacokinetic (PK/PPK) studies) comparing EI/CI versus STI in children with severe infection. There was no limitation on language, and a manual search was also conducted. The data were screened, evaluated, extracted, and reviewed by two researchers independently. Quantitative (meta-analysis) or qualitative analyses of the included studies were performed. Twenty studies (including two RCTs, one case series, six case reports, and 11 PK/PPK studies) were included in this review (CRD42020162845). The RCTs’ quality evaluation results revealed a risk of selection and concealment bias. Qualitative analysis of RCTs demonstrated that, compared with STI, an EI (3 to 4 h) of meropenem in late-onset neonatal sepsis could improve the clinical effectiveness and microbial clearance rates, and reduce the rates of mortality; however, the differences in the incidence of other adverse events were not statistically significant. Retrospective studies showed that children undergoing an EI of meropenem experienced satisfactory clinical improvement. In addition, the results of the PK/PPK study showed that an EI (3 or 4 h)/CI of carbapenems in severely infected children was associated with a more satisfactory goal achievement rate (probability of target attainment) and a cumulative fraction of response than STI therapy. In summary, the EI/CI of carbapenems in children with severe infection has a relatively sufficient PK or pharmacodynamic (PD) basis and satisfactory efficacy and safety. However, due to the limited quantity and quality of studies, the EI/CI therapy should not be used routinely in severely infected children. This conclusion should be further verified by more high-quality controlled clinical trials or observational studies based on PK/PD theories.

## 1. Introduction

Carbapenems are atypical β-lactam antibiotics with a broad antibacterial spectrum, strong antibacterial activity, high stability to β-lactamase, and satisfactory safety profile [1], and include imipenem, meropenem (MER), ertapenem, biapenem, panipenem, and doripenem. They possess good antibacterial activity against most Gram-positive bacteria, Gram-negative bacteria, anaerobic bacteria, and multi-drug-resistant (MDR) bacteria [2]. Owing to the limited selection of antibacterial drugs, carbapenems have gradually become an empirical or target treatment option for severe pediatric infections, such as sepsis, bacterial meningitis, pneumonia, intra-abdominal infections, bone and joint infections, urinary tract infections, and particularly, MDR infections [3,4].

Carbapenems have a time-dependent antibacterial activity; that is, the antibacterial effect is positively correlated with the time at which free concentrations remain above the minimum inhibitory concentration (MIC) as a function of the dosing interval (%fT > MIC). For nonsevere infections, the %fT > MIC required for carbapenems to achieve antibacterial activity was 40%. For severe infections, higher levels need to be considered (fT > MIC, 70 to 80%, or even higher). Therefore, when the maximum dose of an antibacterial drug is used, it is often necessary to extend the infusion time (3 to 4 h or continuous infusion [CI]) to achieve better pharmacokinetic (PK) and pharmacodynamic (PD) targets [5]. Short-term infusion (STI) (0.5 h) in the standard treatment plan often fails to achieve a therapeutic effect. When children of different ages are severely infected, pathological or iatrogenic pathophysiological changes may occur, which may affect the PK process of these drugs in the body [6]. Therefore, specific optimization strategies for the extended infusion (EI) or CI administration of carbapenems in children with severe infections could be an urgent clinical problem to be solved.

At present, there is no clinical practice guideline, expert consensus, or systematic review that provides clear recommendations for the use of carbapenems via EI/CI for severe pediatric infections. Therefore, this systematic review aimed to summarize the current literature on the EI/CI of carbapenems compared with STI administration in children with severe infections.

## 2. Results

### 2.1. Description of Included Studies

Of the 834 records screened, 20 were eligible for inclusion in the final analysis, consisting of two randomized clinical trials (RCTs) (222 patients), one case series study, six case reports (four of which also conducted PK analyses), and 11 PK/population PK (PPK) studies (Table 1 and Table 2 and Figure 1). One of the two included RCTs was adjudged to be at high risk of bias for blinding because the trial was open-label [7]. The other trial was adjudged to be at high risk for sequence generation and was evaluated as having unclear risk in allocation concealment and blinding owing to the lack of information [8] (Appendix A). Publication bias was not assessed in the two included RCTs. The Grading of Recommendations Assessment, Development and Evaluation approach (GRADE) was not performed, owing to the insufficient data.

### 2.2. RCTs

Two RCTs [7,8] (222 cases) reported on the efficacy and safety of late-onset sepsis (LOS) after the EI of MER. Due to the clinical heterogeneity between the studies and the small sample size, only a descriptive analysis was conducted.

Shabban et al. [7] compared the clinical and microbiological efficacy and safety of the EI (over 4 h) versus STI (0.5 h) of MER in 102 infants with Gram-negative LOS, at dosing regimens of 20 mg/kg/dose (q8 h) and 40 mg/kg/dose (q8 h) in meningitis and *Pseudomonas* infection, respectively. The results demonstrated that the EI group showed a significantly higher rate of clinical improvement (61.0% vs. 33.0%, p = 0.009) and microbiologic eradication (82.0% vs. 56.8%, p = 0.009) one week after treatment compared to the STI group. The mortality rate (14.0% vs. 31.0%, p = 0.03), duration of respiratory support (12.5 d [5.7–17.2] vs. 4 d [0–18], p = 0.03), change in C-reactive protein levels (12 mg/L [6–96] vs. 72 mg/L [6–100], p = 0.01), and acute kidney injury rate (6.0% vs. 23.5%, p = 0.02) were significantly lower in the EI group than those in the STI group. However, there was no significant difference in the length of stay in the neonatal intensive care unit (27 d [13–43] vs. 23 d [14–33], p = 0.31), duration of mechanical ventilation (1 d [0–10] vs. 3 d [0–8], p = 0.60), and the rates of other adverse events (AEs) between the groups.

Wang et al. [8] reported on the efficacy and safety of the EI of MER (3 h) compared with the STI of MER (0.5 h) in 120 neonates with LOS, at 20 mg/kg/dose (q8 h). The results showed that the duration of clinical symptom remission (3.1 ± 1.8 d vs. 6.2 ± 1.6 d, *p* = 0.036), the mortality rate (0.0% vs. 6.7%, *p* = 0.042), mechanical ventilation rate (8.1% vs. 21.7%, *p* = 0.041), and levels of infectious biomarkers (white blood cell count, proportion of neutrophils, C-reactive protein levels, and procalcitonin levels) were significantly lower in the EI group than those in the STI group. There was no significant difference in safety outcomes between the two groups.

### 2.3. Observational Studies

One case series [9] and six case reports [10,11,12,13,14,15] revealed the management experience of the EI/CI of MER in critically infected children. Lu et al. [9] reported that 16 of 24 children with acute leukemia and agranulocytosis with nosocomial infections experienced clinical improvement upon undergoing the EI of MER (20 mg/kg/dose, q8 h, over 3 h). Only one patient experienced the loss of appetite. Saito et al. [10] reported the case of a 19-month-old patient treated with extracorporeal membrane oxygenation and continuous hemodialysis, who tested positive for extended-spectrum β-lactamase-producing *Escherichia coli* (MIC ≤ 1 mg/mL, susceptible to MER), and showed negative blood cultures following dose escalation from 120 mg/kg/dose to 300 mg/kg/dose (q8 h, 3 h) of the EI of MER. However, the patient died due to multiple organ failure. In addition, five case reports [11,12,13,14,15] showed that children with severe infections recovered when the dosages were increased or the mode of infusion was changed from STI to CI. Three cases [11,12,14] achieved a probability of target attainment (PTA) of 100% after adjustment.

### 2.4. PK/PPK Data

PPK models or Monte Carlo simulations were generated to assess the outcomes of the EI/CI of carbapenems in children with severe infections in 11 studies. Except for one study involving imipenem, MER was the object of the remaining studies. The following is a descriptive analysis based on infusion time, and a summary of PTA and the cumulative fraction of response (CFR) based on MICs, dosages of administration, and infusion times in each study (only studies with comparative data before and after EI/CI administrations were included). The PTA increased with the increases in dose and infusion time, and with the decrease in MICs (Table 3).

For the 3 h EI route, Kongthavonsakul et al. [16] compared the PTA of intravenous bolus and a 3 h infusion of MER (20 mg/kg q8 h) in severely infected children. The results showed that for organisms with MICs of 1 mg/L, intravenous bolus injection could not achieve the target PTA, while an EI time of 3 h improved the PTA. Courter et al. [17] calculated the PTA and CFR of imipenem/cilastatin (15/25 mg/kg q6 h) and MER (20/40 mg/kg q8 h) using administration times of 3 h and 0.5 h in 2- and 12-year-old children with bacterial infections. The results indicated that the 0.5 h infusion could not yield an ideal PTA at a MIC of 8 mg/L (breakpoint), whereas the 3 h EI route significantly improved the PTA and CFR of both imipenem and MER. Pettit et al. [18] also confirmed the efficacy of the 3 h EI of MER. Simulations were conducted in children with pulmonary exacerbation, and the PTA was calculated and compared between routine 0.5 h and 3 h infusions of MER (40 mg/kg q8 h) for the treatment of *Pseudomonas aeruginosa* infections at different MICs. The results suggested that the PTA for the conventional 0.5 h infusion at a MIC of 1 mg/L was 87.6%. At MICs of 2 and 4 mg/L, the PTAs were below 80%, the 3 h infusions increased PTAs to more than 99% at these MICs, and the PTAs could still reach 82.8% at 8 mg/L with good tolerability. In addition, Hassan et al. [19] built PPK models using data pooled from published studies. The researchers also found that MICs of 2 and 4 mg/L and an EI time of 3 h yielded higher PTAs than conventional 0.5 h infusions did.

The 4 h EI route was mostly utilized among younger children. Anker et al. [20] established a PPK model of MER in preterm and term infants. The most appropriate dosing interval was 8 h. At MICs of 4–8 mg/L, a 40 mg/kg dosage and 4 h EI route achieved a more satisfactory PTA compared with the STI (0.5 h). Padari et al. [21] compared the effect of EI (4 h) and STI (0.5 h) on the PK of MER (20 mg/kg q12 h) in very-low-birth-weight neonates (gestational age < 32 weeks, birth weight < 1200 g). The noncompartmental PK analysis showed that, except for higher peak serum concentrations and the shorter time to peak concentrations in the STI group than in the EI group, the rest of the PK parameters were similar (8/10 in the EI group, and all patients in the conventional infusion group achieved an %fT > MIC of 100% at a MIC of 2 mg/L). A MIC of 6.2 (value required to prevent resistance development in *Pseudomonas aeruginosa*) was reached in a similar proportion (0.5 h: 80.2% vs. 4 h: 81.9%). Therefore, at a MIC of 2 mg/L, 0.5 h administrations in very-low-birth-weight neonates seemed to be acceptable compared with the EI route. In addition, Cies et al. [22] established a PPK model of MER in children with severe infection and compared the PTA difference between MER (40 mg/kg q8 h) with 4 h EI and STI (0.5 h). At MICs ≤ 4 mg/L, STI could provide optimal PTA (≥90%), whereas when MICs ranged from 4 to 8 mg/L, the 4 h EI route could yield an optimal PTA.

Four studies explored the feasibility of the CI of MER in children with severe infections. Rapp et al. [23] reported that the CI of MER (60 mg/kg/d or 120 mg/kg/d) at a higher MIC (>4 mg/L) was effective in patients with normal or augmented renal clearance who underwent continuous renal replacement therapy without drug accumulation. Cojutti et al. [24] suggested that at a MIC of 2 mg/L, the CI of MER (15 to 60 mg/kg/d) could achieve the target of steady-state concentration or MIC ≥ 4 and PTA ≥ 90% in severely infected children with creatinine clearances of 40–300 mL/min/1.73 m^2^. At a MIC of 8 mg/L, the CI of MER (90 mg/kg/d) might achieve an optimal PTA in children with creatinine clearances of 40–120 mL/min/1.73 m^2^. In addition, Wang et al. [25] found that the PTA values for pathogens with high MICs (4 and 8 mg/L), such as *Enterobacteriaceae* and *Pseudomonas aeruginosa*, could still reach 98.0% and 73.3%, respectively, in severely infected infants and children after CI of MER (110 mg/kg/d). Germovsek et al. [26] performed simulations based on a PPK model in neonates and infants with severe infections and stated that the EI/CI modes could increase the plasma PTA but might decrease the PTA in cerebrospinal fluid. Therefore, increasing the dosage or frequency of administration should be considered carefully.

## 3. Discussion

With the increasing number of enterobacteria producing extended-spectrum beta-lactamases and MDR pathogens, carbapenems are often used as “last-line agents” for the treatment of clinically complex and severe bacterial infections [27]. Severe drug resistance, and particularly, the emergence of carbapenem-resistant strains, has forced us to rethink how to optimize the dosing regimens of carbapenems by EI/CI administration based on the PK/PD characteristics of such drugs for severe infections, to achieve a better PK/PD target [28]. There is no clinical practice guideline that provides specific recommendations for the application of the EI/CI of carbapenems in children with severe infections. The present review is the first systematic review to narratively synthesize the feasibility of the EI/CI of carbapenems in children with severe infections in RCTs, case reports, and PK/PPK studies. The results demonstrated that the optimized dosing regimens of carbapenems have a sufficient PK/PD theoretical basis, which could achieve a more satisfactory PTA, and several successful cases of treatments have been reported. However, owing to the limitations of sample size and quality of RCTs, there is insufficient evidence to support the routine use of the EI/CI of carbapenems as a standard regimen among children with severe infections.

The optimal use of carbapenems in both adults and children with severe infections is a significant and controversial issue. For adult patients, the EI/CI of carbapenems was reported to improve the rates of clinical effectiveness and bacterial eradication [29] and the mortality rate [30,31]. For pediatric patients, severe infections are often accompanied by a series of pathophysiological changes, while conventional dosing regimens may result in difficulty reaching the standard PK/PD targets, leading to treatment failures. In addition to increasing the dose of drugs to the maximum, extending the infusion time of carbapenems may result in better PK/PD targets [32], which is also confirmed in other β-lactam drugs [33].

PPK studies are recognized as appropriate tools to determine the dosage and usage optimization in children with complex conditions. Traditional PK studies of children are often limited by blood sample volumes and time, poor compliance, and potential ethical issues [34]. PPK modeling can use the probability of distribution of PK parameters after children are given a limited dose of drugs. It optimizes the design of clinical trials and dosing regimens by simulating the PK/PD target attainments under different dosing regimens [35,36]. The 11 PK/PPK studies and four case-based PK studies in this review all suggested that the EI/CI of carbapenems could improve or achieve satisfactory PTAs (mainly an fT > MIC of 40%), which might be associated with better clinical outcomes. However, in addition to the infusion time, the dosing intervals, dosages, and MIC values of different bacteria may also affect the therapeutic endpoints. When the MIC is low (≤2 mg/L), STI administration can achieve the target PTA, whereas when the MIC is high (≥4 mg/L), it may be necessary to increase the dose or extend the infusion time to achieve satisfactory PTAs.

There still remains a number of controversies and challenging unresolved issues. First, the stability of carbapenems is poor [5]. MER should not be infused continuously for 24 h. It can maintain stability for 7 h and 5 h at 22 °C and 33 °C, respectively [37], while doripenem (5 mg/mL in normal saline) can remain stable at room temperature for 12 h [38]. It is necessary to cold-pack the septum of the infusion pumps or increase the frequency of replacing the infusion bag and dosing devices, which may increase the cost of care and the risk of treatment failures. Second, carbapenems generally have good water solubility and are mostly excreted through the kidneys as prototypes [1]. In cases of renal insufficiency (or end-stage renal disease being treated with continuous renal replacement therapy), the half-life will be prolonged and drug accumulation should be considered cautiously. Furthermore, although the EI/CI modes can increase the PK/PD targets in the plasma, they may reduce the %fT > MIC in cerebrospinal fluid [24] and urine [39]. In addition, in very-low-birth-weight infants with severe infections, EI/CI routes have no significant PK/PD advantages over STI [21]. Therefore, further PK/PPK studies are needed to explore whether the EI/CI of carbapenems can achieve treatment goals regarding nonbloodstream infections, such as meningitis and complex urinary tract infections. Finally, except for MER, which is approved for the treatment of complex pelvic infections in children of all ages, there is still a substantial off-label use of carbapenems among infants and newborns younger than 3 months. 

The present review has certain limitations. First, most of the included studies focused on MER and only one involved imipenem. Therefore, the conclusions may not be extrapolated to other carbapenems. Second, in addition to optimizing the dosing interval, the dosage, interval, and local drug resistance of carbapenems are also significant factors in the therapeutic management; however, this was not highlighted as a key component of our review. Therefore, clinical decision-making needs to integrate these factors. Finally, owing to the limited quality and insufficient data from clinical trials and observational studies, we could not conduct meta-analyses, but a narrative synthesis in this systematic review.

## 4. Materials and Methods

The present systematic review followed the guidelines of the preferred reporting items for systematic review and meta-analyses (PRISMA) checklist [40], as shown in Appendix A, with a protocol registered with PROSPERO (CRD42020162845).

### 4.1. Eligibility Criteria

We included studies of children with severe infections that compared the EI/CI to STI of carbapenems, regardless of publication types, mainly comprising clinical practice guidelines, systematic reviews or meta-analyses, RCTs, cohort studies, case-control trials, case reports, and PK/PD studies.

We identified the incidences of clinical improvement, mortality, and serious AEs as main outcomes. The secondary outcomes were the incidences of microbial elimination and AEs, length of hospital stay, respiratory support time, and infectious biomarkers. For PK/PPK studies, the PTA and CFR values were considered as outcomes. We excluded republished studies or conference abstracts, letters, and errata that did not provide sufficient information and data.

### 4.2. Search Strategy

We searched the PubMed, Embase, the Cochrane Library, Clinicaltrials.gov, China National Knowledge Infrastructure, WanFang Data, and SinoMed databases from their inceptions to 10 August 2020, using keywords, MeSH terms, and Emtree headings, including “meropenem,” “imipenem,” “biapenem,” “panipenem,” or “carbapenems,” and routes of administration with pediatric filters (Appendix A). There was no language limitation in this study. In addition, we manually searched the reference lists of included studies, reviewed the literature, and contacted the authors for missing or unpublished data, if necessary. 

### 4.3. Data Extraction

Pairs of reviewers (Z.P.X. and Z.Y.H.) independently screened titles, abstracts, and full-text articles, and then extracted data using a standardized form. Disagreements were resolved by discussion or a third investigator (Z.S.D.). The reviewers collected basic information on the characteristics of the included articles (first author, year of publication, study design, and target drugs), patient characteristics (country, age, disease, pathogen, number of patients, treatments, and assessment time), and outcomes of interest. For PK/PD studies, we additionally extracted the number of samples, dosage regimen (dose, dosing interval, and infusion time), number of simulations, dosage regimen (dose, dosing interval, and infusion time), and PK/PD target. 

### 4.4. Quality Assessment

For each eligible article, two reviewers (Z.P.X. and W.Z.H.) independently assessed the risk of bias for RCTs using the Cochrane Risk of Bias tool and assessed the quality of systematic reviews or meta-analyses using the AMSTAR (A Measurement Tool to Assess Systematic Reviews) version 2 [41]. We also evaluated the overall quality of evidence for each outcome using the GRADE system, if possible. Any disagreements were resolved by discussion or a third reviewer (Z.S.D.).

### 4.5. Statistical Analysis

Meta-analyses were performed using Review Manager version 5.3 (The Nordic Cochrane Centre, Copenhagen, Denmark) if there were more than two RCTs. Binary variables were reported as risk ratios, while continuous variables were presented as mean differences with 95% confidence intervals. Heterogeneity between studies was calculated using the Cochrane *Q* statistic (χ^2^  *p* values) and the *I*^2^ statistic of inconsistency. The fixed-effects method was used if the I^2^ was less than 25% (*p* ≥ 0.1), and the random-effects method was used if the I^2^ was 25% or greater (*p* < 0.1). For high-heterogeneity pooled outcomes, we conducted subgroup analyses according to control groups, dosage, MIC values, etc. Sensitivity analyses were performed by changing the statistical methods and models. Analysis items with a two-tailed *p* < 0.05 were considered statistically significant. Funnel plots were used to assess the possibility of publication bias.

A narrative synthesis was provided if a meta-analysis was inappropriate. We only performed qualitative descriptive analyses according to research types.

## 5. Conclusions

The present systematic review showed that the application of the EI/CI of carbapenems among children with severe infections has a relatively sufficient PK/PD theoretical basis and limited clinical data on efficacy and safety. Therefore, these modes of administration cannot yet be used as routine regimens among children with severe infections. The EI/CI of carbapenems may be used with caution only when serious infections caused by drug-resistant bacteria or strains with high MIC values are suspected. In the future, high-quality controlled clinical trials or observational studies with sufficient sample sizes based on PK/PD theories are needed to address this significant issue.

## Figures and Tables

**Figure 1 antibiotics-10-01088-f001:**
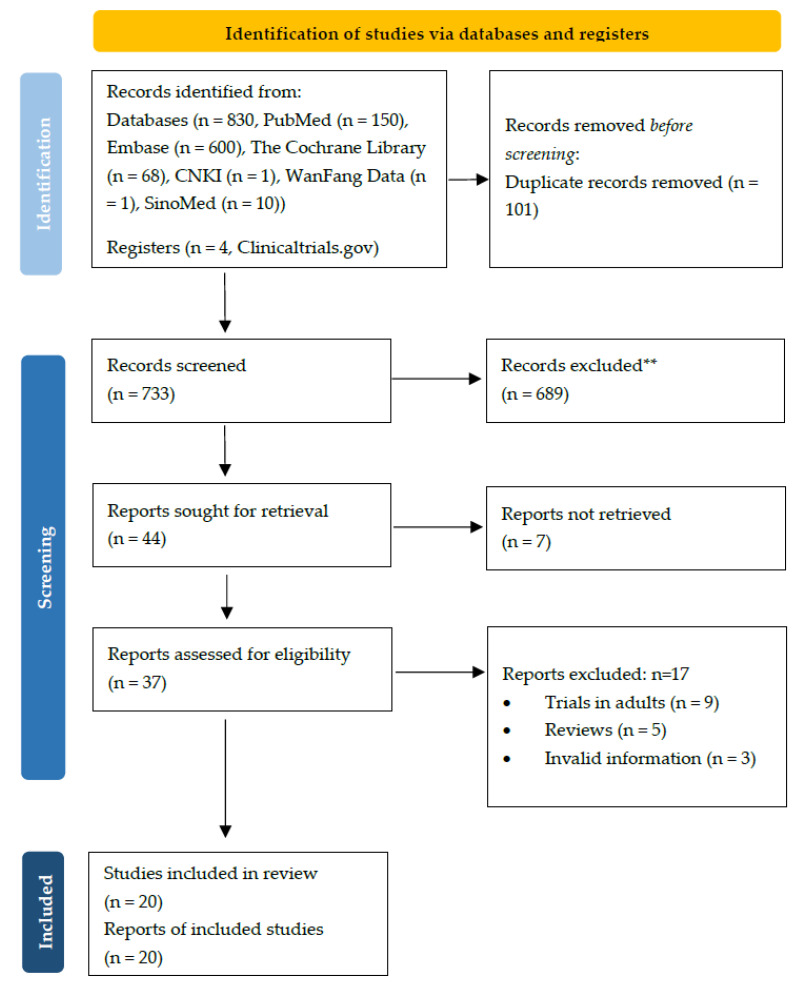
The PRISMA flow diagram.

**Table 1 antibiotics-10-01088-t001:** Basic information of included RCTs and retrospective studies.

Reference	Study Design	Country	Disease and Pathogen	Age (T/C)	Sample Size (T/C)	Dosage Regimen (Doses, Dosing Interval and Infusion Time)	Efficacy Outcomes and Evaluation Time	Safety Outcomes
Shabaan 2017 [7]	RCT	Egypt	Gram-negative late-onset sepsis; most frequently *Klebsiell* a species and *E. coli*	33.5 w ± 3.8/ 34.3 w ± 3.5	102 (51/51)	meningitis infection: 20 mg/kg/dose q8 h; *Pseudomonas* infection: 40 mg/kg/dose q8 h T: MER > 4 h C: MER 0.5 h	primary outcome: clinical improvement rate; microbiologic eradication rate (D7) secondary outcomes: mortality; meropenem-related (MR) duration of mechanical ventilation; MR length of NICU stay; total length of NICU stay; duration of respiratory support; duration of mechanical ventilation, MR duration of inotropes; CRP concentrations (D7)	incidence of MR-AEs (D7): AKI, diarrhea, rash, seizures, nausea and vomiting and increased levels of liver transaminases
Wang 2018 [8]	RCT	China	LOS; NA	30.3 w ± 2.1/ 30.5 ± 3.1	120 (60/60)	MER 20 mg/kg/dose q8 h T: 3 h C: 0.5 h	WBC count, N%, levels of CRP and PCT, duration of clinical symptoms remission; mortality; rate of assisted ventilation treatment	incidence of AKI, levels of ALT, AST, urea nitrogen and creatinine, skin rash and gastrointestinal symptoms
Lu 2010 [9]	Case series	China	nosocomial infections with leukemia and agranulocytosis; blood culture positive in 4 cases *	9 mo~15 y (mean age: 5.4 y)	41	MER 20 mg/kg/dose q8 h over 3 h	clinical effective rate (D3)	incidence of AEs
Saito 2020 [10]	case report	Japan	ECMO with continuous hemodialysis for postoperative management; bacteremia; ESBL-*E. coli*	19 mo	1	MER POD 2, 120 mg/kg/d q8 h 3 h; after two infusions, 200 mg/kg/d q8 h 3 h; POD 5, 300 mg/kg/d q8 h 3 h	blood culture result, clinical outcome	NA
Cies-1 2014 [11]	Case report	United states	ECLS-ECMO; *Pseudomonas* *aeruginosa;* MIC = 0.5 μg/mL	8 mo	1	MER D13 40 mg/kg 0.5 h, followed by 200 mg/kg/d, CI	serum MER concentration (8 h, D3 and D9), blood culture result at D13–21 of ECLS or decannulation, PTA	NA
Cies-2 2014 [12]	Case report	United states	MIC = 0.25 μg/mL ECLS-CRRT, sepsis, *Pseudomonas aeruginosa*; MIC = 0.25 mcg/mL	10 d	1	MER D12 40 mg/kg 0.5 h, followed by 240 mg/kg/d CI	serum MER concentration (D12 and D15), blood culture result (D13–15), PTA	NA
Falagas 2006 [13]	Case report	Greece	post-appendectomy septicaemia; *Klebsiella pneumoniae*; MIC = 8 mg/L (intermediately susceptible)	15 y	1	MER D9-D10 1 g q8 h ; D11–D17 2 g q8 h; D18 6 g/d CI	temperature, WBC count; continuous evaluation	NA
Cies 2015 [14]	Case report	United states	ventriculitis; *Serratia marcescens;* MIC ≤ 0.25 μg/mL	2 y	1	MER D26 40 mg/kg/dose q6 h over 0.5 h; D27 200 mg/kg/d CI	serum and CSF MER concentration, PTA	NA
Zobell 2014 [15]	Case report	United states	cystic fibrosis, MDR *Inquilinus limosus;* MIC = 4 mcg/mL	13 y	1	MER 500 mg (51/mg/d) q8 h, 3 g–6 g 22.5~23.5 h CI, multiple hospitalization	improvement of pulmonary function test	blood counts, renal function, and hepatic function

Note: RCT, randomized controlled trial; MER, meropenem; LOS, late-onset sepsis; *E. coli*, *Escherichia coli*; ESBL— *E.coli*, extended-spectrum β-lactamase-producing Escherichia coli; MDR, multi-drug resistant; POD, postoperative day; CI: continuous infusion; EI: extended infusion; STI: short-term infusion; WBC, white blood cell; N%, proportion of neutrophil granulocyte; CRP, C-reactive protein; PCT, procalcitonin; AKI, acute kidney injury; ALT, alanine transaminase; AST, aspartate aminotransferase; ECMO, extracorporeal membrane oxygenation; ECLS, extra-corporeal life support; CRRT, continuous renal replacement therapy; PTA, probability of target attainment; CSF, cerebrospinal fluid; AEs, adverse events; NA, not accessible. * *Pseudomonas Aeruginosa*, *Staphylococcus Epidermidis*, *Escherichia Coli*, and *Candida Pelliculosa.*

**Table 2 antibiotics-10-01088-t002:** Basic information of included PK/PPK studies.

References	Study Design	PK/PPK Studies			Monte-Carlo Simulation			
		Subject	Number of Patients/Blood Samples	Dosage Regimen (Doses, Dosing Interval and Infusion Time)	Number of Simulations	Dosage Regimen (Doses, Dosing Interval, and Infusion Time)	MIC (mg/L)	PK/PD Target
Bolus vs. 0.5 h~1 h vs. 3 h EI								
Kongthavonsakul 2016 [16]	PPK modeling and simulation	children with severe infection	14/84	MER 20 mg/kg/dose q8 h i.v. bolus	10,000	MER 20, 30, 40 mg/kg/dose q8 h i.v. bolus/1 h/3 h EI	1, 2, 4, 8, 16	40% fT > MIC
Courter 2009 [17]	simulation	children with bacterial meningitis or other infections	99/425	MER 10, 20, 40 mg/kg single dose over 5 min or 0.5 h imipenem/cilastatini: 25 mg/kg 15–20 min q6 h	5000	MER 20/40 mg/kg/dose q8 h 0.5 h/3 h EI imipenem/cilastatin: 15/25 mg/kg q6 h 0.5 h/3 h EI	0.016–128	40% fT > MIC, PTA ≥ 90%, CFR ≥ 90%
Pettit 2016 [18]	PPK modeling and simulation	children with CF hospitalized for an acute pulmonary exacerbation	30/120	MER 40 mg/kg/dose q8 h 3 h	5000	MER 40 mg/kg/dose q8 h 0.5 h/3 h EI	0.03–128	40% fT > MIC
Hassan 2020 ^#^ [19]	PPK modeling and simulation	children with infections	288/NA	MER 10–40 mg/kg/dose q8 h–q12 h	1000	MER < 50 kg: 20 mg/kg/dose q8 h 0.5 h; 40 mg/kg/dose q8 h 0.5 h; 20 mg/kg/dose q6 h 0.5 h; 20 mg/kg/dose q8 h 3 h EI > 50 kg: 1 g/dose q8 h 0.5 h; 2 g/dose q8 h 0.5 h; 1 g/dose q6 h 0.5 h; 1 g/dose q8 h 3 h EI	2, 4	40% fT > MIC
0.5 h vs. 4 h EI								
Anker 2009 [20]	PPK modeling and simulation	pre-term and full-term neonates	38/342	MER 10, 20, 40 mg/kg single dose 0.5 h	10000	MER 20/40 mg/kg/dose q8 h/q12 h 0.5 h/4 h EI	Up to 8	40% fT > MIC
Padari 2012 [21]	PPK model verification	very-low-birth-weight neonates	19/114	MER 20 mg/kg/dose q12 h 0.5/4 h	NA	NA	2	fT > MIC; fT > 6.2 MIC
Cies 2014 [22]	PPK modeling and simulation	children in PICU	11/(2–3 per child)	MER standard doses *	1000	MER 40 mg/kg;q8 h; 0.5 h/4 h EI	0.25–32	40% fT > MIC;PTA > 90%
CI								
Rapp 2020 [23]	PPK modeling and simulation	children in PICU	40/121	MER 20 mg/kg/dose q8 h over 20 min/CI	400	MER 20 mg/kg q8 h 20 min/3 h EI; 40 mg/kg q8 h 20 min; 60 mg/kg/d CI; 120 mg/kg/d CI	Up to 8 mg/L	50% fT > MIC; 100% fT > MIC
Cojutti 2015 [24]	Retrospective PK study and simulation	underwent hematopoietic stem cell transplantation with suspected or documented Gram-negative infection	21/44	MER Mean 92.69 mg/kg/d CI	10,000	MER 15, 30, 45, 60, 90 mg/kg/d CI	0.25–256	Css/MIC ≥ 1 and ≥ 4, PTA ≥ 90%
Wang 2020 [25]	PPK modeling and simulation	critically ill infants and children (bacterial meningitis, sepsis, severe pneumonia)	57/135	MER Mean 23.7 ± 8.59 mg/kg, most 0.5–1 h	100	MER 20–60 mg/kg/dose; q6 h/q8 h/q12 h; 2/4 h infusion, CI	1, 2, 4, 8	70% fT > MIC, PTA70%
Germovask 2018 [26]	PPK modeling and simulation	neonates and infants (sepsis, or bacterial meningitis, or pleocytosis, or a positive Gram-stain from the CSF)	167/401	MER meningitis: 40 mg/kg; sepsis: 20 mg/kg <32 week’s GA and <2 week’s PNA q12 h, others q8 h > 0.5 h	1000	MER 20/40 mg/kg bolus/CI	0.25, 0.5, 1, 2, 4, 8, 16	Plasma (for sepsis) or CSF (for meningitis) 61%T > MIC

^#^ PPK modeling based on three published research data; * the specific doses were unknown; PPK, population pharmacokinetic; i.v., intravenous; CF, cystic fibrosis; PICU, pediatric intensive care units; CSF, cerebrospinal fluid; CRRT, continuous renal replacement therapy; GA, gestational age; PNA, postnatal age; MIC, minimum inhibitory concentration; PTA, probability of target attainment; CFR, cumulative fractions of response; Css, steady-state concentration; NA, not accessible; PICU, pediatric intensive care units; EI, extended infusion; CI, continuous infusion.

**Table 3 antibiotics-10-01088-t003:** PTA and CFR under different optimization schemes of carbapenems.

MIC (mg/L)	Dosage (mg/kg)	References	PTA 40 % fT > MIC				
			Bolus	0.5 h STI	1 h EI	3 h EI	4 h EI
MER							
1	20	Courter 2009		92.0		100.0	
		Kongthavonsakul 2016	67.8			100.0	
		Saito 2020			46.3	63.8	
	40	Courter 2009		98.0		100.0	
		Cies 2014		98.0			100.0
		Pettit 2016		87.6		>99.0	
		Saito 2020			60.0	76.3	
	66.67	Saito 2020			100.0	100.0	
	100	Saito 2020			100.0	100.0	
2	20	Courter 2009		72.0		100.0	
		Kongthavonsakul 2016	40.0		67.3	100.0	
		Hassan 2020		68.4		>97.5	
		Padari 2012		100.0			100.0
	40	Courter 2009		92.0		100.0	
		Cies 2014		96.0			100.0
		Pettit 2016		70.1		>99.0	
4	20	Courter 2009		33.0		97.0	
		Kongthavonsakul 2016	15.5			99.9	
		Hassan 2020		41.7		90.7	
	40	Courter 2009		72.0		100.0	
		Cies 2014		90.0			100.0
		Pettit 2016		35.4		>99.0	
8	20	Courter 2009		3.0		54.0	
	40	Courter 2009		33.0		97.0	
		Cies 2014		71.5			99.6
		Pettit 2016		10.0		82.8	
CFR (%)							
NA	20	Courter 2009		91.0		95.0	
				84.0		98.0	
	40			94.0		98.0	
				93.0		98.0	
Imipenem and Cilastatin Sodium							
1	60	Courter 2009		58.0		100.0	
	100			66.0		100.0	
2	60			45.0		95.0	
	100			55.0		99.0	
4	60			31.0		74.0	
	100			41.0		91.0	
8	60			18.0		38.0	
	100			27.0		65.0	

Note: The darker the color, the higher the PTA value. MER, meropenem; MIC, minimum inhibitory concentration; PTA, probability of target attainment; CFR, cumulative fractions of response; 40% fT > MIC, Time > MIC of free drug meets or exceeds 40% of the dosing interval.

## Data Availability

The data presented in this study are available in Appendix A. The unpublished data retrieved by private correspondence with the authors and used in this study analysis are available on request from the corresponding authors.

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
