# Peer review of "Extended or Continuous Infusion of Carbapenems in Children with Severe Infections: A Systematic Review and Narrative Synthesis"

_antibiotics, 2021, doi:10.3390/antibiotics10091088_

Round 1
Reviewer 1 Report
All comments have been adressed.
Author Response
Thank you so much for your kind comments and approval.
Best wishes,
Xiaomei Tong, Suodi Zhai, Pengxiang Zhou
on behalf of the coauthors of this manuscript.
Reviewer 2 Report
In this review entitled "Extended or Continuous Infusion of Carbapenems in Children with Severe Infections: A Systematic Review and Narrative Synthesis" the authors have systematically reviewed the efficacy and safety of extended or continuous infusion (EI/CI) versus short-term infusion (STI) of carbapenems in children with severe infections. Qualitative and quantitative analysis were performed from data obtained from different studies available from databases. But, due limited quantity and quality of studies, the authors conclude that the EI/CI therapy should not be used routinely in severely infected children and must be further verified by more high-quality controlled clinical trials or observational studies based on PK/PD theories.
Major concerns:
- Why the authors didnt explain and/or suggest the PK/PD theories which they think are suitable for this work?
- The authros mentioned in the abstract about more high quality controlled clinical trials? what does the author wants to suggest from this statement? Are clinical trials not of high quality or are their standards different from country to country?
- The search strategy implemented is upto August 10, 2020? One year ago? Can authors clarify about the period of analysis?
- The records screened and records excluded show a huge number of variation. Almost 94 percent being ignored! How the exlcuded records may not be significantly important for this study? How just 20 included studies be representative of Carbapenems related issue discussed by the authors? Cant they explain any obsevational studies or theroies which authors wants to conclude!
- What are high-quality controlled clinical trials as per authors? Can they define the parameters in a tabular form to make this study effective?
Minor concerns:
- What is the rationale for this study? How authors define it from their experince?
- I think Moderate English changes required!
Author Response
Dear reviewer,
We gratefully thank you for your precious time making constructive remarks and useful suggestions, which has significantly improved the quality of the manuscript and has enabled us to improve the manuscript. Each suggested revision and comment was seriously considered and revised to the most extend. Please see the following explanations and resubmitted manuscript with revision model.
Major concerns
Q1: The PK/PD theories of EI/CI of carbapenems were already described in the background (from line 52 to 60), briefly summarizing the core basis of this kind of administration optimization in children with severe infections. We didn't further discuss the relative theories because the PK/PPK studies were not in priority compared to clinical studies in this systematic review but could provide theoretical feasibility of EI/CI in children.
Q2: Owing to the included clinical trials were of moderate quality with a high risk of blinding and allocation (Table S3), we reckoned that the current RCTs were insufficient to strong conclusions. To our knowledge, not all of the published or registered clinical trials are high quality as we imaged. It is important to evaluate the RCTs using tools (like Cochrane risk of bias tools in this review) to reveal the potential biases, which could be significant for us to conduct appropriate analyses. Therefore, we need further RCTs or observational studies with low bias and sufficient samples to verify the conclusions of this review.
Q3: For the search implementation, we conducted the literature search on August 10th, 2020, and completed the record screening and selection in November, and then finished the data analyses in January 2021. From February to March, we wrote the manuscript. Finally, we submit it to Antibiotics after language improvements. However, we were encouraged to resubmitted the manuscript after revising in light of reviewers' comments the first time, which took us approximately 2 months. However, after conducting a supplementary search from August 11th, 2020 to September 4th, 2021 in databases, we found that there were no additional studies further influencing the conclusions. Therefore, we thought the deadline for the literature searching might be appropriate.
Q4:
The whole process of literature searching, screening, and extraction fully followed the PRISMA checklist and Cochrane handbook, with a protocol registered with PROSPERO. The reason why the majority of original records were ignored in the end was that we conducted a sensitivity-maximizing search in databases, only consisting of drugs and the pediatric population. So we could include target studies as many as possible. Additionally, the exact reasons for records removal in secondary screening were summarized in Figure 1, while the main reasons in primary screening included irrelevant studies, trials in adults, or reviews, which were not required to be displayed in Figure 1.
As for the representativeness of 20 studies, we had no restriction on certain carbapenems (please see 4.2 or Table S2) in the search strategies. However, as a result, most of the included studies focused on meropenem, and only one involved imipenem, which could be a limitation and was also mentioned in the manuscript. Additionally, we couldn't find any observational controlled studies, such as cohort studies or case-control studies up to now. So, the results of this review tend to represent the currently available evidence and conclusion of this significant issue in pediatric patients. Our conclusions didn't recommend the routine use of EI/CI administration of carbapenems among children owing to the insufficient data of clinical trials.
Q5: As we mentioned in Q2, high-quality clinical controlled trials are at the top of the evidence pyramid of intervention trials, so we were eager to find these kinds of evidence to illustrate the target clinical questions. Nevertheless, not all controlled trials were designed or conducted with no bias, so both Cochrane Library and PRISMA suggested reviewers use checklists to objectively evaluate the potential bias of trials and analyze with caution accordingly. We performed the risk of bias evaluation of two included randomized controlled trials in the manuscript (Table S3), which were adjudged to be at high risk of bias for blinding and sequence generation. Therefore, the current support from included trials was insufficient for suggesting the routine use of EI/CI routes, and further high-quality trials with sufficient samples should be conducted in this field.
Minor concerns
Q1: The systematic review has been a widely recognized method to comprehensively collect, summarize or analyze the current best available evidence to deal with a particular clinical issue. It's the best study type for medical staff to make decisions through evidence-based ways. Like this systematic review and narrative analysis recommended, pediatric physicians could refer to the conclusions and conduct EI/CI administration of carbapenems with caution only when serious infections caused by drug-resistant bacteria or strains with high MIC values are suspected. Also, they could further conduct clinical trials based on this review to fill in gaps of this crucial issue.
As for the definition of the questions from our experience, we found the uncertainty of EI/CI routes of carbapenems in children with severe infections through clinical practice initially and then decided to conduct this systematic review together with pediatric physicians. We promoted clinical pathways of carbapenems use in severely infected children according to our review and decided to administrate the EI route with caution, in consideration of infection sites, resistance, and MIC values.
Q2: We have invited native speakers to rechecked the whole manuscript and tried our utmost to modified the grammar and improve the expressions. Please see the resubmitted manuscript, and the revised parts were already highlighted with red color.
Again, thank you so much for your kind comments and valuable suggestions. Please let us know if more revisions are needed.

This manuscript is a resubmission of an earlier submission. The following is a list of the peer review reports and author responses from that submission.
Round 1
Reviewer 1 Report
line 66- consensus does not have a pleural
line 82- singular tense
and throughout the text, there are a few errors, but overall this article is fine. thank you for asking me to review
Reviewer 2 Report
Dear authors,
In my oppinion there is insufficient data to presented in this systematic review due to lack of published studies. The Methods seem to be generic as they mention analysis when there are more than two RCTs, what is not the case here. Furthermore, Figure 1 seems to be missing what is unimaginable for this kind of paper.
In the present paper, the title is too general and does not adequately describe the question the authors wanted to address. Research in the area of pediatric application of antimicrobials is called for. However, in the present paper I cannot confirm that it has the expected value of a systematic review as it does not have analysis consistent with systematic reviews. There simply isn't enough published research for such analysis or the search criteria were inadequate. They were previously published in a protocol not taking into consideration does such search retrieve results. Moreover, the Manuscript body is poorly written as the Abstract stresses the methodology rather than the results and Methods are generic but do not in any way, other than last two sentences refer to the Manuscript. Moreover, figure 1 mentioned in both Manuscript and Checklist seems to be missing. Outstanding methodology should be encouraged, however, pointless secondary studies should not. Should we ask the authors to publish another systematic review when there is one more RCT published again? The overall Manuscript and the writing style appears to be more of a narrative review than a systematic review. I would encourage authors to actually do pooled analysis of the studies in question and not just narrate their results. Moreover, exact methodology on included and excluded studies (Figure 1) is missing and hence the Manuscript cannot be considered for publication in the present form.
Reviewer 3 Report
This is a timely and relevant review of evidence on role of carbapenems for severe infections in children. Superfluous abbreviations make the worthwhile read unnecessarily daunting. A textbox key to abbreviations could be a helpful addition, but paring out some of the least-used and thus less justified abbreviations would enhance readability without greatly increasing the page length.
- Pharmacotherapeutic options are limited for serious infections in pediatric patients. Carbapenems comprise a subclass of beta-lactam antibacterial drugs with broad spectrum relative to other beta-lactams, all parenteral, and time-dependent antibacterial activity. For serious infections, extended and continuous infusion to maintain inhibitory/bactericidal concentrations at the locus of infection may be required to clear infection without recrudescence. Guidance for protracted use of carbapenems and target kinetics in treating serious infections in pediatric patients is lacking. Carbapenems are potentially less toxicity than other classes of broad-spectrum drugs for serious infections, for example less oto-vestibulo-nephro-toxic than aminoglycosides for aerobic Gram-negative infections, and to avoid potential tendinopathy-neurotoxic and cardiac conduction effects associated with fluoroquinolones. Meanwhile, a systematic synthesis of current evidence to inform clinical practice is lacking. This systematic review was undertaken to take stock of the current body of evidence concerning use of carbapenems as a therapeutic option for children with serious infections.
- The originality and relevance are that systematic synthesis of current evidence to inform clinical practice is lacking. This systematic review was undertaken to take stock of the current body of evidence concerning use of carbapenems as a therapeutic option for children with serious infections.
- The authors established that a systematic synthesis of current evidence to inform clinical practice was lacking. To minimize bias (selection, ascertainment, reporting), they gathered and critiqued existing literature using a disciplined approach to take stock of the current body of evidence concerning use of carbapenems as a therapeutic option for children with serious infections. That is the point of a systematic review.
- The authors used the PRISMA checklist to provide an explicit transparent replicable literature search strategy, and guide and score risks of bias in their approach to inclusion/exclusion and analysis of literature in this systematic review.
- The authors conclude that there remains insufficient sound clinical data to support a standard approach for grading of evidence (GRADE), and insufficient sound clinical data to back up the putatively rational pharmacological basis for maintaining supra-MIC concentrations for extended periods of treatment of otherwise treatment-resistant infections in children, so pending needed generation of more compelling clinical evidence, that merits proportionate caution when carbapenems are used accordingly.
6.The references are appropriate,current and apparently comprehensive.
- Figure 1 illustrates application of inclusion/exclusion criteria to studies retrieved and considered for review. Tables S1...S3 pertain to PRISMA components.. standard for systematic review.
Overall, this is a well written account of the currently rather deficient body of current clinical evidence on relative safety and effectiveness of maintaining supra-MIC concentrations for extended periods of treatment in pediatric cases of otherwise treatment-resistant infections.
Reviewer 4 Report
Zhou P, et al have submitted a systematic review to compare extended or continuous infusion of carbapenems to short term infusion in children with severe infections. The paper analyzes a very hot-topic in time of MDRO infection especially in pediatric/neonatal population. Unfortunately there is still a lot to do, and many other data need to be produced before recommending with strength the EI/CI of carbapenem, as authors have correctly claimed.
I have no suggestions to further improve the work.